# Machine Learning Approaches for Ship Speed Prediction towards Energy Efficient Shipping

**Misganaw Abebe [1] , Yongwoo Shin [2] , Yoojeong Noh [2,* ], Sangbong Lee [3] and Inwon Lee [4]**

1    Research Institute of Mechanical Technology, Pusan Nat'l University, Busan 46241, Korea;
misge98@pusan.ac.kr
2    School of Mechanical Engineering, Pusan Nat'l University, Busan 46241, Korea; ds_vitamin@naver.com
3    Lab021, Busan 48508, Korea; sblee@lab021.co.kr
4    Department of Naval Architecture & Ocean Engineering, Pusan Nat'l University, Busan 46241, Korea;
inwon@pusan.ac.kr
*    Correspondence: yoonoh@pusan.ac.kr; Tel.: +82-51-510-2308

**Abstract:** As oil prices continue to rise internationally, shipping costs are also increasing rapidly. In order to reduce fuel costs, an economical shipping route must be determined by accurately predicting the estimated arrival time of ships. A common method in the evaluation of ship speed involves computing the total resistance of a ship using theoretical analysis; however, using theoretical equations cannot be applied for most ships under various operating conditions. In this study, a machine learning approach was proposed to predict ship speed over the ground using the automatic identification system (AIS) and noon-report maritime weather data. To train and validate the developed model, the AIS and marine weather data of the seventy-six vessels for a period one year were used. The model accuracy result shows that the proposed data-driven model has a satisfactory capability to predict the ship speed based on the chosen features.

**Keywords:** ship speed over the ground; machine learning; ship fuel consumption; decision tree regression; ensemble methods

## 1. Introduction

Due to the increase in oil prices, shipping industries have been struggling to reduce fuel expenses. According to Stopford [1], the cost of fuel oil consumption is nearly two-thirds of the overall voyage costs and more than one-fourth of the total running costs of a ship. Because of this, shipping industries have been striving to employ measures for fuel efficiency. Based on previous studies that examined the route planning of ships, it was found that the economic efficiency of a ship can be managed by choosing a suitable route with a consideration of the sea state (weather data) [2]. To find the proper ship route, an accurate prediction of ship speed is necessary. Previous studies showed that ship speed can be estimated by evaluating ship speed loss based on its resistance.

The total resistance of a ship can be obtained from the summation of resistance due to wind and waves, the rudder effect, drift, water temperature, surface pressure, and salinity. The resistance of a ship can also be estimated using analytical or numerical methods. Roh [3] proposed a method of finding an economical shipping route to reduce fuel expenses by considering the resistance of a ship using analytical equations from ISO 15016 (ISO, 2015) [4]. Kim et al. [5] estimate ship speed loss using both 2-D and 3-D potential flow methods and computational fluid dynamics with an unsteady Reynolds-averaged Navier–Stokes approach. They also compared simulation results with analytical approaches to the ship resistance in calm water and with added resistance due to wind and irregular waves corresponding to the Beaufort scale.

However, due to the difficulty in modelling actual sea surfaces and estimating the total energy system of a ship, inaccuracies are anticipated in the calculated results. To overcome this problem, a data-driven model has been proposed. Yoo and Kim [6] investigated ship performance in terms of ship speed and engine power using the Gaussian process and polynomial regression models for single container ship data. Gan et al. [7] proposed an algorithm to build an improved multilayer perceptron network for predicting long-term ship speed, by applying particle swarm optimization to optimize the hidden neurons of the multilayer perceptron. Lui et al. [8] conducted a comparison study on a recurrent neural network, back propagation neural network, and support vector regression model to investigate the trajectory of a ship using a single ship in a certain area using automatic identification system (AIS) sensor data.

Choosing the right machine learning model to evaluate ship speed and assess ship performance while sailing is always challenging, especially when applied to big data [9–11]. Applying a simple model, such as a linear regression, may not be precise enough [12]. In addition, it might be difficult to determine the features necessary to train the model and to tune the hyperparameters [13,14]. This study proposes a maritime data analysis framework based on AIS and marine weather data to predict ship speed over the ground (SOG), which determines the most economical shipping route that can reduce fuel expenses. This framework includes data acquisition, preprocessing such as denoising, feature extraction, and model generation. To generate the model for SOG, various machine learning regression techniques are employed, such as, linear regression (LR), polynomial regression, decision tree regressors (DTRs), gradient boosting regressors (GBRs), extreme gradient boosting regressors (XGBRs), random forest regressors (RFRs), and extra trees regressors (ETRs) where their parameters are optimized through hyperparameter tuning. Using real ship route data, the computational time and accuracy of each method were compared through model validation, and the most accurate and efficient method was validated for various ship routes and ship types. The developed methodology in this study is expected to be used to train the best models for the SOG prediction of ships, which aims to track the performance of ships, and finally be used for actual ship route optimization purposes.

The remaining sections of this study are organized as follows. Section 2 describes the suggested methodology which includes data pre-processing, formulation of the regression models, parameter tuning methods, and model verifications. Section 3 explains the details of the case study and offers a discussion of the results. Section 4 provides the overall conclusions of the study.

## 2. Material and Methodology

This section provides the details of data acquisition, a proper pre-processing method, and feature selection for the given dataset. Details are also provided of the development and implementation of various models following various modelling methodologies, the optimization of the hyperparameter of the potential models, and finally, a comparison of the models to determine the most efficient modelling method. A graphical depiction of the developed methodology is shown in Figure 1.

### 2.1. Data Acquisition

A 2018 AIS satellite data and noon-report weather data of 14 tanker and 62 cargo ships were collected. The AIS data and noon-report marine weather data was provided by Lab021, and the AIS data collected within an average of 3 min time intervals. The resolution of the weather data is 0.5 degrees in the latitude and longitude directions. In this study, the proposed framework was validated by using five datasets with different types of routes and ships among the total data. The description of both the AIS and weather data is shown in Table 1. The AIS data consist of static information, dynamic information, and navigation information. Static information includes the identification numbers of the ship such as its Maritime Mobile Service Identity (MMSI) and International Maritime Organization (IMO) number, call sign and name, ship's types and dimensions (dimension A-D), and location of the electronic fixing device antenna. Since static information is rarely changed, the data are manually updated. Dynamic data include operational information related to the navigation of a ship. The data

are collected with some time interval (data time stamp) and automatically updated according to the navigational status of the ship.

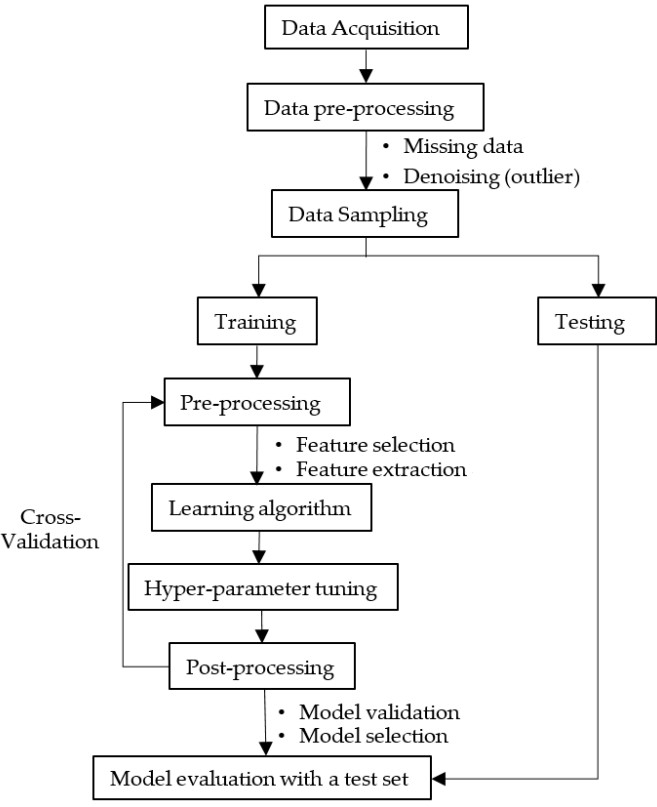

**Figure 1.** Scheme of the suggested methodology.

**Table 1.** Parameters of the automatic identification system (AIS) and weather data.

| No | Parameter | Unit | Remark | No | Parameter | Unit | Remark |
|---|---|---|---|---|---|---|---|
| 1 | MMSI | – | | 24 | Total wave height | m | |
| 2 | IMO number | – | | 25 | Total wave direction | deg. | |
| 3 | Call sign | – | | 26 | Total wave period | sec | |
| 4 | Name | – | | 27 | Wind wave height | m | |
| 5 | Type of ship | – | AIS data for static info (Msg. type 5) | 28 | Wind wave direction | deg. | |
| 6~9 | Dimension A~D | m | | 29 | Wind wave period | sec | |
| 10 | Electronic fixing device | – | | 30 | Swell wave height | m | |
| 11 | ETA | sec | | 31 | Swell wave direction | deg. | |
| 12 | Max draught | m | | 32 | Swell wave period | sec | Weather data (0.5° resolution) |
| 13 | Msg type | – | | 33 | Wind UV | m/s | |
| 14 | Date time stamp | KST | | 34 | Wind VV | m/s | |
| 15 | MMSI | – | | 35 | Mean sea pressure level | hPa | |
| 16 | Latitude | DMS | | 36 | Pressure surface | hPa | |
| 17 | Longitude | DMS | | 37 | Ambient temperature | °C | |
| 18 | SOG | knot | AIS data for dynamic info (Msg. type 123) | 38 | Sea surface salinity | Psu | |
| 19 | ROT | deg/min | | 39 | Sea surface temperature | °C | |
| 20 | COG | deg. | | 40 | Current UV | m/s | |
| 21 | True heading | deg. | | 41 | Current VV | m/s | |
| 22 | Navigational status | – | | | | | |
| 23 | Msg type | – | | | | | |

Similarly, weather data considerably affect ship speed, such that it must include main features to predict the performance of a ship (SOG). For example, as the hull of a vessel goes to the sea, it highly induces resistance when it is sailing due to friction and wave-making [15]. Frictional resistance only

occurs at the drawn part of the body, and thus, the loading condition of the ship and the roughness of the hull have an effect on the hull resistance of a vessel. Waves also cause additional resistance due to the pitch and heave motions of the vessel and the reflection of the short waves on the hull [16]. Since wave resistance is continually varying over time and will be added to the total resistance obtained at each specific location, it must be considered when predicting the performance of a ship over its voyage. The total resistance of a vessel also depends on the properties of the water and is directly proportional to the viscosity and density of the seawater [15]. A higher viscosity or a higher density of the water will increase the resistance of the vessel. The viscosity and density of seawater depend on the salinity and temperature of the water, which may change based on the body of water, location, and period of the year. Likewise, the relative ship speed such as SOG is highly dependent on the ocean current [17]. Based on the studies of Chen [17] and Calvent [18], ship heading and speed are influenced by the ocean current; these studies also suggested that the actual SOG is the vector summation of the current and heading direction where UV and VV are the speeds for the longitudinal axis (u-axis) and lateral axis (v-axis) directions of the earth, correspondingly. If the ocean current movement comes from the heading direction, the ship sailing will be against it; but if it is in the opposite direction, the ocean current will increase the SOG of the ship.

Among all features, some features related to the static and dynamic information are single-valued and non-numeric features that have no effect on the results and are removed. The remaining features are listed in Table 2, which also includes the length, width, gross-tonnage, and deadweight of the ship which may have an effect on ship speed based on the information mentioned above. Next, because of the potential existence of missing and outlier data, the identification of anomalies and undesirable data points and pre-processing are needed in the following dataset acquisition stage.

**Table 2.** Chosen features.

| Remark | No. | Features | Units |
|---|---|---|---|
| | 1 | Max draught | m |
| | 2 | Course over the ground (COG) | deg. |
| | 3 | True heading | deg. |
| | 4 | Total wave height | m |
| | 5 | Total wave direction | deg. |
| | 6 | Total wave period | sec |
| | 7 | Wind wave height | m |
| | 8 | Wind wave direction | deg. |
| | 9 | Wind wave period | sec |
| | 10 | Swell wave height | m |
| | 11 | Swell wave direction | deg. |
| | 12 | Swell wave period | sec |
| | 13 | Wind UV | m/sec |
| Input Features | 14 | Wind VV | m/sec |
| | 15 | Pressure at mean sea level (MSL) | hPa |
| | 16 | Pressure surface | hPa |
| | 17 | Ambient temperature | °C |
| | 18 | Sea surface salinity | Psu |
| | 19 | Sea surface temperature | °C |
| | 20 | Current UV | m/s |
| | 21 | Current VV | m/s |
| | 22 | Ship length | m |
| | 23 | Ship width | m |
| | 24 | Dead weight | tons |
| | 25 | Gross tonnage | tons |
| Output | 1 | SOG | knots |

## 2.2. Data Preprocessing

In this section, the pre-processing for the acquired dataset is presented.

1.  To investigate only the operating periods of the ship, this study extracted the "Underway using engine" data from the Navigational Status features, which meant the mooring and anchoring periods were rejected.
2.  Shipping speed can decrease due to different sea state resistances; however, there is also a probability that it may be reduced by the operator, especially around the port at the start and end of the voyage. To reduce this kind of measurement error, this study discarded the data with less than 5 knots of SOG, which is considered as maneuvering.
3.  From the AIS data report [19], if the data value is not-available (missed data), there is a default outlier value for each feature such as 102.2 for SOG, 511 for heading, 91 for latitude and 181 for longitude [20]. Those values were observed in our data and used to discard the missed data.
4.  The scatter plot of the features can be used to show that the data may have noise/outliers because of the inconsistencies in the measurement of the sensors or human errors which must be rejected before training the models. Z-score is a parametric outlier detection method for different numbers of dimensional feature space [21]. However, this method assumed that the data had a Gaussian distribution; hence, the outliers were considered to be distributed at the tails of the distribution, which meant that the data point was far from the mean value. Before deciding a threshold that we set as $Z_{thr}$, the given data point $x_i$ was normalized as $Z_i$ using the following equation.

$$Z_i = \frac{x_i - \mu}{\sigma},\tag{1}$$

where μ and $\sigma$ are the mean and standard deviation of all $x_i's$, respectively. An outlier is then a data point that has an absolute value greater than or equal to $Z_{thr}$:$|Z_i| \geq Z_{thr}$.

Usually, the threshold value is set to ±3 [22]; however, our data is extremely non-linear and this study only aims to remove extreme cases. Therefore, the study used a threshold value of ±5 for all features to reject values which were extremely far from the mean value on both tails. Figures 2 and 3 show examples of the data distribution of SOG, including normal and outlier data, which are detected using the Z-score.

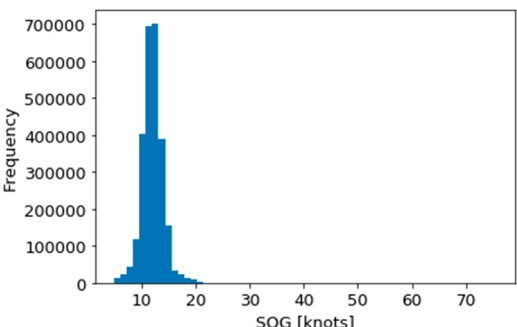

**Figure 2.** Speed over the ground (SOG) distribution.

## 2.3. Feature Selection and Extraction

Feature selection was needed to remove unnecessary features. Before feature selection was conducted, some features were converted to a more convenient format. For example, the data for wind and current were collected in vector form, but for convenience, it was converted to a scalar form that still well captures the information enclosed in the dataset. Wind and current speed was converted to its magnitude and direction angles where the magnitude of speed was obtained using the equation $|\mathbf{V}| = \frac{\pi}{180} \times \sqrt{u^2 + v^2}$, and the direction was calculated using $\theta = 180 + \frac{180}{\pi} atan2(u, v)$.

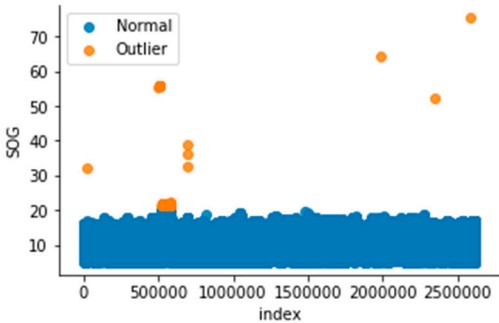

**Figure 3.** SOG scatter plot.

To remove unnecessary features, a high correlation filter was conducted. The definition of high correlation filter [23] in this study is that, if the observed values of two input features are always the same, it means that they represent the same entity. Thus, highly correlated variables are considered as one variable. The result of the correlation matrix of 25 input features is shown in Figure 4. Pairs of features with a correlation coefficient higher than 0.7 were taken as one, thereby reducing the number of input features to 13. The acquired weather data of total wave (height, direction, and period) was obtained from the square roots of the sum of wind and swell (height, direction, and period), and thus, they were expected to have a high correlation. Since the ship COG is the actual direction of the vessels, it is highly correlated with true heading. Gross tonnage is calculated by multiplying length, width, and breadth, and thus, it is highly correlated the dimensions of the ship and with the dead weight, which is the weight of everything aboard the ship. The final selected features are shown in Table 3, as mentioned in Section 2.1 all the chosen features have an effect on the ship speed performance while the ship is sailing.

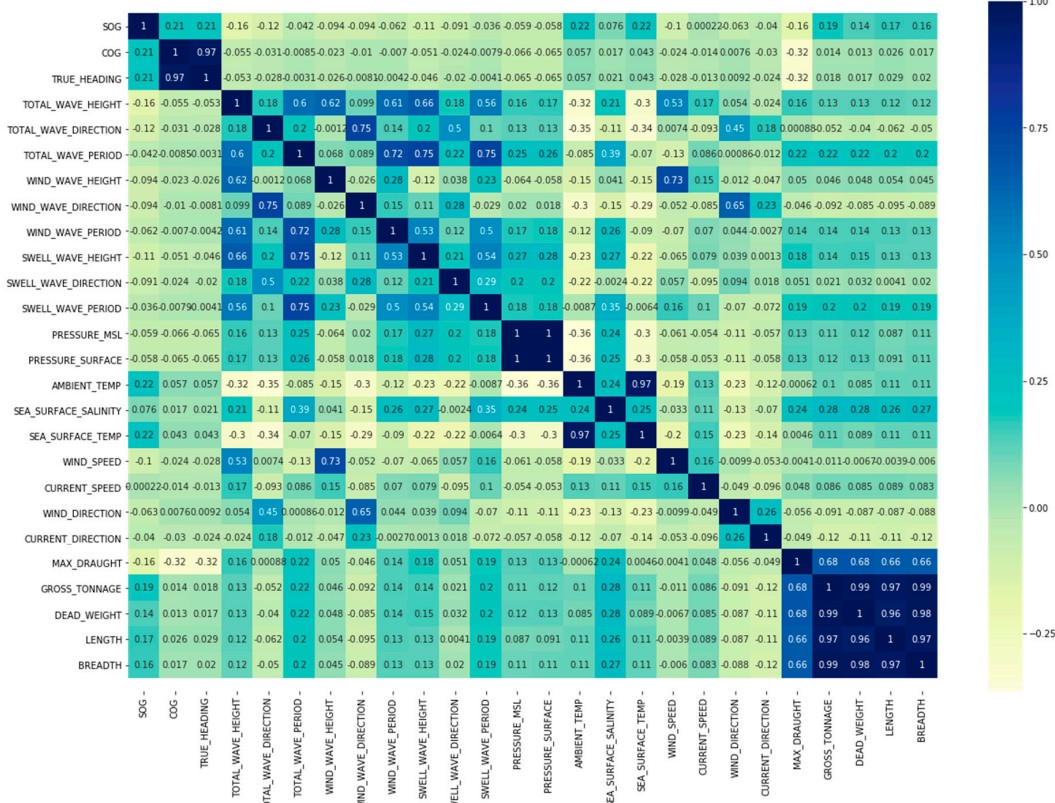

**Figure 4.** Feature correlation matrix.

**Table 3.** Final selected features.

| Remark | No. | Features | Units |
|---|---|---|---|
| | 1 | Max draught | m |
| | 2 | COG | deg. |
| | 3 | Total wave height | m |
| | 4 | Total wave direction | deg. |
| | 5 | Total wave period | sec |
| Input Features | 6 | Wind speed | m/sec |
| | 7 | Wind speed | m/sec |
| | 8 | Pressure MSL | hPa |
| | 9 | Ambient temperature | °C |
| | 10 | Sea surface salinity | Psu |
| | 11 | Current speed | m/s |
| | 12 | Current speed | m/s |
| | 13 | Gross tonnage | tons |
| Output | 1 | SOG | knots |

## 2.4. Prediction Models

The SOG value of a ship involves environmental disturbance, which is difficult to model using conventional parametric approaches. In view of this complexity, this section describes the modeling techniques and the general method followed in this study to build potential machine learning models for ship speed prediction, such as DTR, and ensemble models, such as GBR, XGBR, RFR, and ETR.

### 2.4.1. Decision Tree Regressor

DTR is a non-parametric supervised learning regression method [24] in the form of a tree structure with nodes and branches. In DTR, the features are partitioned into a rectangle space and a simple model (tree) is trained for each feature. The models are learned using a training dataset on a continuous range. Their output ends up being the mean value of the observations of training sets that are located on the same node. Classification and Regression Trees (CART) is one of the most common methods for tree-based regression methods. In CART, the feature space will be split into two regions after choosing the optimal split point to obtain the best model fit. This will execute recursively until the stopping rules are activated.

To develop the model, for a given $n$ number of dataset samples and $d$ number of features, $\mathcal{D}\{(\mathbf{x}_i, y_i)\} \left( |\mathcal{D}| = n, \ \mathbf{x}_i \in \mathbb{R}^d, \ y_i \in \mathbb{R} \right)$ the feature space is assumed to be split into $K$-number of regions, called $R_K$ and the prediction value of the model is obtained from the average value of the observation which lies in the $k^{\text{th}}$ region:

$$\hat{y}_i = \boldsymbol{ave}(y_i | \mathbf{x}_i \in R_k). \tag{2}$$

The best $\hat{y}_i$ can be obtained by minimizing the least square error of $\sum \left( y_i - \overline{y}_i \right)^2$. Though optimal $\hat{y}_i$ values can be simply calculated, however, it is not easy to split the region. To overcome this, a greedy algorithm is applied recursively to determine the optimal splitting nodes until the stopping point is triggered. Usually, this depends on the hyperparameters and the difficulty of the fundamental problem. The selectable hyperparameters are:

- The maximum depth of the tree (*max_depth*) which indicates how deep the built tree can be. The deeper the tree, the more splits it has, and it captures more information about the data; however, increasing depth could increase the computation time.

- *min_samples_split* represents the minimum number of samples required to split an internal node. This can vary between considering only one sample at each node to considering all of the samples at each node. When this parameter is increased, the tree becomes more constrained as it has to consider more samples at each node.
- The minimum number of samples required to exist at each leaf node (*min_samples_leaf*). This is similar to *min_samples_splits*, however, this describes the minimum number of samples at the leaves.
- In addition, the number of features (*max_features*) to consider while searching for the best split should be specified.

### 2.4.2. Ensemble Methods

The basic idea of the ensemble learning method is developing a prediction model by integrating a number of simple models. The two most common ensemble learning methods are boosting [25] and bagging [26].

Bagging is a method that integrates several individual base models into one to generate an inclusive ensemble model. A new prediction model can be developed from separate prediction models to form an ensemble, for instance, by averaging regression. Averaging individual models means reducing the variance; and thus, bagging can be applied for a model with high variance and low bias. As opposed to bagging, boosting is a common method to generate an ensemble model from a single model such as, decision trees. It is a sequential technique that integrates a set of weak learners and provides a more accurate model estimation. The boosting model produces strong models with low bias. The new outcomes of the developed model have weights based on the earlier outputs of the model. If the outputs are predicted properly, a smaller weight is assigned; otherwise, the assigned weight will be higher.

*Random Forest Regressor*

RFR has been proposed by Breiman [27] and was developed based on the bagging technique. To construct the RFR model, a number of decorrelated decision tree regressors (*n_estimators*) are generated using the presented training dataset. The response of the RFR model is considered by averaging the outcomes of individual decision trees:

$$\hat{y}_i(\mathbf{x}) = \frac{1}{M} \sum_{m=1}^{M} f_m(\mathbf{x}_i) \tag{3}$$

where $M$ is the number of decision trees (*n_estimators*). To construct a decision tree, the method uses a bootstrap replica of the training sample and the CART algorithm. An optimal split over a subsample of features at each test node is obtained by searching a random subsample with the size of the contender features. This means that a subsample without replacement is selected from the contender features with the smallest sample size to split the node. In scikit-learn implementation, similar to DTRs, the minimum number of samples required to split an internal node is controlled by a *min_samples_split* parameter.

*Extra Trees Regressor*

The ETR algorithm develops an ensemble of unpruned regression trees based on the standard top-down process. The difference between ETR and RFR is that the selected cut-points of the split nodes in ETR are extremely random to grow the tree, in addition, ETR uses the whole training sample instead of a bootstrap replica [28].

As for its numerical features, the splitting procedure of ETR has two basic parameters, which are the *number of features* randomly chosen at each node and the *minimum sample size* for splitting a node. To obtain the final result, ETR formulates the predictive models of the individual trees, as in

RFR, and the predicted models are combined to produce the final prediction result, such as averaging in regression problems. The basic hyperparameters are the *number of features* to govern the strength of the feature selection procedure, the *minimum sample size* to strengthen the averaging of the outcome noise, and the *number of trees* to strengthen the variance reduction of the ensemble model combination.

In a scikit-learn implementation, the hyperparameters are similar to those of DTR, with additional information about the number of trees (*n_estimators)* in the forest. Usually, a higher number of trees better trains the data. However, adding a lot of trees can slow down the training process considerably, therefore a parametric search to find the optimal configuration is necessary.

*Gradient Boosting Regressor*

GBRs are based on the boosting meta-algorithm, which yields an estimation model in the form of an ensemble of weak prediction models usually using decision trees [29]. GBRs construct an additive model in a stage-wise fashion, and it allows the optimization of arbitrary differentiable loss functions. To formulate the GBRs a tree ensemble model uses *M* additive functions to estimate the output.

$$\hat{y}_i(\mathbf{x}) = \sum_{m=1}^{M} f_m(\mathbf{x}_i), \ \ f_m \in \mathcal{F} \tag{4}$$

where $\mathcal{F}$ denotes the function space which includes the whole regression trees $\mathcal{F} = \left\{ f(\mathbf{x}) = w_{q(\mathbf{x})}, \ w \in \mathbb{R}^T, \ q : \ \mathbb{R}^d \to T \right\}$. $q$ denotes the structure of each tree that maps the corresponding leaf index. $T$ denotes the number of leaves in the tree. Each $f_m$ corresponds to an independent tree structure $q$ and leaf weight $w$. Unlike DTRs, each regression tree contains a continuous score on each leaf, and here, $w_j$ represents the score on the $j^{th}$ leaf. The leaf weight is calculated by minimizing the loss function:

$$\mathcal{L} = \sum_{i} l(\hat{y}_i, y_i) + \frac{1}{2}\lambda \sum_{j=1}^{T} w_j^2, \tag{5}$$

where, *l* represents a differentiable loss function that measures the difference between the prediction $\hat{y}_i$ and the target $y_i$. $\lambda$ denotes a regularization constant value to penalizes the complexity of the model, and the optimal $w_j$ can be obtained using a second-order Taylor series approximation of Equation (6) [30].

$$w_j = \frac{\sum_{i \in I_j} \frac{\partial l(y_i, \hat{y}_i)}{\partial(\hat{y}_i = 0)}}{\sum_{i \in I_j} \left( \frac{\partial^2 l(y_i, \hat{y}_i)}{\partial(\hat{y}_i = 0)^2} \right) + \lambda}, \tag{6}$$

where $I_j$ is a dataset contained at a leaf *j*.

In scikit-learning implementation, a GBR also has the same main hyperparameters as a DTR with the addition of *n_estimotors* and *learning _rate* which may help the model shrink the contribution of each tree.

*Extreme Gradient Boosting Regressor*

XGBRs are an optimized distributed GBR, which are designed to be efficient, flexible, and portable [31]. XGBR provides additional regularization hyperparameters as shown in Equation (7), which can help reduce the chances of overfitting, decrease prediction variability and, therefore, improve accuracy. The predicted output $\hat{y}_i$ is obtained by minimizing the regulation function $\mathcal{L}$:

$$\mathcal{L} = \sum_{i} l(\hat{y}_i, y_i) + \sum_{m} \Omega(f_m) , \ where \ \Omega(f) = \gamma T + \frac{1}{2}\lambda \|w\|^2 + \alpha|w| \tag{7}$$

Here, $\Omega$ represents the regularization parameter that penalizes the complexity of the model like regression tree functions and smooth the final learned weights to avoid overfitting. *T* represents the number of leaf nodes and *w* is the score of the leaf node. $\gamma$, $\lambda$, and $\alpha$ are used to define the level of

regularization. $\alpha$ and $\lambda$ also known as $L_1$ and $L_2$ regularization, respectively, have different influences on weight; $\alpha$ inspires sparsity, encouraging the weight to be zero, while $\lambda$ inspires the weight to be small. $\gamma$ is a commonly implemented pseudo-regularization hyperparameter known as a Lagrangian multiplier which controls the complexity of a given tree. $\gamma$ specifies the minimum loss reduction required to make further partitions on a leaf node, which means that a higher value leads to fewer splits. In addition to the use of a regularization term, predictor *subsampling* was used to prevent overfitting [30].

The prediction process adds the results of each tree to obtain the final results in the XGBR model. The parameters of each tree ($f_t$), which includes the structure of the tree and the scores obtained by each leaf node, have to be determined. The additive training method adds the result of a tree to the model at a given time. The predicted value ($\hat{y}_i^{(t)}$) obtained in step $t$ can be used to obtain the algorithm process:

$$\hat{y}_i^{(t)} = \sum_{m=1}^{M} f_m(\mathbf{x}_i) = \hat{y}_i^{(t-1)} + f_t(\mathbf{x}_i) \tag{8}$$

In a scikit-learn implementation, the additional parameters in GBRs are $\gamma$, $\lambda$, and $\alpha$, as mentioned above. These regularization parameters limit how extreme the weights (or influence) of the leaves in a tree can become.

### 2.5. Model Hyperparameter Tuning

As mentioned in Section 2.4, there are numerous hyperparameters in a model and the change in hyperparameter values can affect the performance of the constructed model. Since the optimal hyperparameter values are not identified at first, optimization should be carried out to select the proper values for each model. The most commonly used method of optimization is the grid searching method [32], which involves all the potential combinations of the chosen hyperparameters and a profound assessment of each hyperparameter to choose the best combination. However, this brings about a substantial cost because of the absolute number of combinations that may have to be evaluated (particularly if the model has several tunable hyperparameters).

Another optimization method is the random search method [32]; in this case, all the hyperparameter ranges are sampled randomly. This method also requires a long-running time because some time may be spent evaluating unpromising areas of the search space.

A model-based method to find the minimum function is called Bayesian optimization [33]. It has lately been used for hyperparameter tuning in machine learning. Bayesian optimization is an algorithm that uses the Bayesian theorem to adaptively generate data for hyperparameters and find the optimum hyperparameter values using surrogate models. It can attain a better performance on a test set with less iterations than a random search or a grid search [10]. To avoid the overfitting of the model and to ensure that the chosen hyperparameter combination values are near the optimal values, a k-fold cross-validation [24] technique is applied. The training dataset was split into k- subsamples, which means that the model will run $k$ times iteratively, using $k-1$ subsamples to train the model and the rest of the subsamples for testing. During each iteration of a combined hyperparameter setup, a number of model accuracy results are obtained and averaged.

### 2.6. Model Validation

To ensure the accuracy of the constructed prediction model, most commonly used error measures such as the coefficient of determination ($R^2$), root mean square error (RMSE), and normalized root mean square error (NRMSE) are used. $R^2$ shows the relative errors of the model fitness and RMSE shows the absolute error of the predicted model. In addition, the NRMSE gives the scale-free RMSE result. The details of model accuracy are provided as follows.

### 2.6.1. Coefficient of Determination ($R^2$)

The $R^2$ is a crucial measurement of model accuracy for regression analysis. It is expressed as the proportion of the variance of the predicted dependent feature to the independent feature. $R^2$ is defined based on the *sum of squares of residuals* ($SS_{res}$) and *sum of squares total ($SS_{tot}$)*. $SS_{res}$ quantifies how far the predicted values of the model are from the observed data, and $SS_{tot}$ quantifies how far the observed data are from the mean value. By changing the combinations of $SS_{tot}$ and $SS_{res}$ values, the constructed regression model can be effectively compared to the mean model. The equations for $SS_{tot}$ and $SS_{res}$ are given as:

$$SS_{tot} = \sum_i \left( y_i - \overline{y} \right)^2$$
$$SS_{res} = \sum_i \left( y_i - \hat{y}_i \right)^2 \tag{9}$$

where $y_i$ is the observed data, $\overline{y}$ is the mean of the observed data, and $\hat{y}_i$ is the predicted value of the model.

The difference between $SS_{tot}$ and $SS_{res}$ estimates the closeness of the regression model compared to the mean model. Dividing their difference by $SS_{tot}$ gives $R^2$ which indicates the goodness of fit of the model. The coefficient of determination defined as:

$$R^2 = 1 - \frac{SS_{res}}{SS_{tot}} \tag{10}$$

The scale of $R^2$ ranges from 0 to 1; 0 indicates that the proposed model does not improve prediction over the mean model, and 1 indicates perfect prediction.

### 2.6.2. Root mean square error (RMSE)

The RMSE is the square root of the variance of the individual differences called residual. It indicates how close the values of the estimation model are with the observed data values. In general, RMSE is an absolute measure of the fitness of a model, while $R^2$ is a relative measure of fitness. A lower value of RMSE denotes a better fit. If the developed model is for prediction purposes, RMSE is an appropriate and accurate measure that can show how the responses are predicted. RMSE is defined as follows:

$$RMSE = \sqrt{\frac{\sum_{i=1}^{n} (y_i - \hat{y}_i)^2}{n}} \tag{11}$$

where $X_o$ represents the observed values and $X_m$ represents the estimated model prediction values at the $i^{th}$ data.

### 2.6.3. Normalized Root Mean Square Error (NRMSE)

NRMSE can be a better measure to evaluate model performance by normalizing the RMSE which can be beneficial by making RMSE scale-free. For example, when converted to a percentage, it is easier to determine the absolute fitness of the prediction model. The normalization of RMSE for a range of observed data is defined as:

$$NRMSE = \frac{RMSE}{y_{i,max} - y_{i,min}} \tag{12}$$

## 3. Model Development

### 3.1. Methodology Application

This section associates the data acquired from 14 tracks and 62 cargo ships and AIS data which contain the dynamic and static data of the journey of the ship and noon-reports of marine weather

data. Using the obtained data, this study aims to evaluate the performance of data-driven regression models in the prediction of SOG.

For the transparency of the built-in studies which are shown in Figure 1, the particular procedure followed to obtain the results was as follows:

1. The acquired dataset was loaded.
2. Unnecessary features such as static information in the AIS data were rejected.
3. Data where the ship is moored and anchored were identified and discarded.
4. Data where the ship has an SOG value of less than 5 knots were discarded.
5. The missed data were identified in the AIS data and discarded.
6. The outliers were discarded for some of the features based on the z-scores.
7. The key features were selected by applying feature selection methods such as a high correlation filter.
8. The dataset was subjected to sampling (splitting) into a training and test set.
9. The models which can potentially estimate the target were listed down.
10. k-fold cross-validation was implemented for each model:

   - Each model was trained using hyperparameter optimization by specifying the range of the search space for each hyperparameter, Bayesian optimization was executed over the specified search space, and the results were assessed.
   - The model was trained using the whole training set after the optimal hyperparameters were obtained.
   - The results of the constructed model results were evaluated using a test set, and the performance metrics were calculated.

11. The constructed models were evaluated using three accuracy measures ($R^2$, RMSE, NRMSE) and overall conclusions were drawn.

### 3.2. Results and Discussion

As explained in Section 2.3, 41 features were reduced to 13 major input features and one SOG output feature, and the results of the descriptive statistics of the dataset after the pre-processing are shown in Table 4.

To verify the validity of the potential models, regression analysis was carried out using the full-scale ship operation data and the regression results using the training and testing dataset were obtained. The classic approach splits the dataset into two randomized sets, those are, the training set and the testing set. The split ratio for the two datasets is between 80/20 and 50/50, depending on how large the dataset is. Here, the dataset was split 67% for training and 33% for testing data.

To clearly understand the relationship between the dependent and independent features, a Pearson correlation analysis [34] was conducted using the training dataset. The correlation coefficient matrix result of SOG to the other measured features is shown in Table 5. The correlation between SOG and the other input factors is not high because the speed of the vessel is determined primarily by the torque and rpm of the vessel; furthermore, other weather-related features have a relatively low correlation. If data on the engines of the ships were collected, highly correlated features could have been included as main features. However, AIS and weather data only are used because the shipping company did not provide such engine data for security reasons, which often occurs in the shipping industry. Thus, AIS and weather-related features are the only ones that can be used to predict the ship's performance. Although only weather and dynamic information are used, the SOG can be still accurately predicted because engine rpm and torque are generally not volatile during the operation of vessels.

**Table 4.** Descriptive statistics of dataset after pre-processing.

| Features | Mean | Std. | Min | 25% | 50% | 75% | Max |
|---|---|---|---|---|---|---|---|
| COG | 172.652 | 98.804 | 0.0000 | 85.100 | 162.800 | 263.100 | 360.000 |
| Total wave height | 2.038 | 0.970 | 0.0002 | 1.421 | 1.964 | 2.573 | 6.759 |
| Total wave direction | 175.432 | 76.611 | 0.2243 | 122.246 | 181.793 | 224.441 | 359.720 |
| Total wave period | 8.371 | 2.293 | 0.8933 | 7.051 | 8.455 | 9.907 | 17.384 |
| Pressure MSL | 1016.53 | 7.08 | 980.06 | 1011.61 | 1016.41 | 1021.10 | 1044.42 |
| Ambient temp | 21.146 | 5.794 | −8.0820 | 17.855 | 21.718 | 25.695 | 36.110 |
| Sea surface salinity | 35.034 | 1.148 | 28.7284 | 34.573 | 35.362 | 35.605 | 41.126 |
| Wind speed | 6.914 | 3.088 | 0.0874 | 4.693 | 6.747 | 8.820 | 22.836 |
| Wind direction | 157.916 | 93.067 | 0.3399 | 90.167 | 134.010 | 230.064 | 359.863 |
| Current speed | 0.318 | 0.225 | 0.0020 | 0.166 | 0.257 | 0.399 | 1.515 |
| Current direction | 160.511 | 89.145 | 0.8856 | 85.932 | 146.343 | 232.631 | 360.000 |
| Maximum draught | 12.747 | 5.278 | 0.0000 | 8.900 | 12.200 | 15.300 | 23.200 |
| Gross tonnage | 93137 | 67667 | 8231 | 38400 | 79560 | 199959 | 200679 |
| SOG | 12.107 | 1.882 | 5.000 | 11.000 | 12.100 | 13.200 | 22.200 |

**Table 5.** Correlation between input features and SOG.

| Features | Correlation Coefficient |
|---|---|
| Ambient temperature | 0.218750 |
| COG | 0.206953 |
| Gross tonnage | 0.192829 |
| Total wave height | 0.161433 |
| Maximum draught | 0.161050 |
| Total wave direction | 0.124137 |
| Wind speed | 0.104488 |
| Sea surface salinity | 0.076264 |
| Wind direction | 0.062621 |
| Total wave period | 0.059107 |
| Pressure MSL | 0.042215 |
| Current direction | 0.039811 |
| Current speed | 0.002221 |

In addition, the result shows that current speed and direction have less effect compared to other features, although the ship SOG is found to be influenced by the ocean current according to the description in Section 2.1. This is because, as shown in Table 4, the maximum current speed in our dataset is 1.515 knots and 75 % of the dataset has less than 0.399 knots, showing that our dataset does not have a highly volatile range of current speed. Nevertheless, because it is known that ocean current affects the performance of a ship, current speed and direction were included as input features to avoid losing the effect of the ocean current.

Next, Bayesian optimization was performed to find the optimal hyperparameter values for each model. The considered hyperparameters for each model and their range of values are given in Table 6 along with the optimal hyperparameter values. The training used a 10-fold cross-validation in order to get a stable result.

To identify the optimal models and hyperparameters, $R^2$ was assessed for each model produced at each fold. To evaluate the post-training of the model, other accuracy measures were computed, as explained in Section 2.6. The result of the model accuracy measurements was also compared with the linear and 3rd order polynomial regression model using the same independent feature sets.

The overview of model accuracy is given in the plots of Figure 5. The three figures on the left indicate the calculated $R^2$ values through 10-fold cross validations using GBR, linear regression, and 3rd order polynomials, respectively, and the figure on the right indicates the ones using XGBR, DTR, RFR, and ETR. The line inside of the box shows the median or second quartile of the model at k-fold, the top and bottom of the box show the first and third quartile, respectively. The whiskers indicated as horizontal lines also show the lowest and the highest points of data within a 1.5 interquartile range of

the lower and upper quartile, respectively. Consequently, data points beyond the whiskers are shown individually as a hollow circle.

**Table 6.** Hyperparameters of models.

| Model | Hyperparameters Tuned | Range | Optimal Value |
|---|---|---|---|
| DTR | max_depth | [1, 100] | 60 |
| | min_samples_split | [2, 10] | 2 |
| | min_samples_leaf | [1, 4] | 1 |
| | max_features | [1, 13] | 6 |
| RFR | n_estimators | [1, 100] | 89 |
| | max_depth | [1, 100] | 50 |
| | min_samples_split | [2, 10] | 2 |
| | min_samples_leaf | [1, 4] | 1 |
| | max_features | [1, 13] | 5 |
| ETR | n_estimators | [1, 100] | 61 |
| | max_depth | [1, 100] | 39 |
| | min_samples_split | [2, 10] | 2 |
| | min_samples_leaf | [1, 4] | 1 |
| | max_features | [1, 13] | 8 |
| GBR | n_estimators | [1, 100] | 50 |
| | learning_rate | [0.01, 1] | 0.1 |
| | max_depth | [1, 50] | 37 |
| | min_samples_split | [2, 10] | 2 |
| | min_samples_leaf | [1, 4] | 1 |
| | max_features | [1, 13] | 7 |
| XGBR | n_estimators | [1, 100] | 57 |
| | learning_rate | [0.01, 1] | 0.2 |
| | max_depth | [1, 50] | 30 |
| | subsample | [0.01, 0.8] | 0.76 |
| | colsample_bytree | [0.01, 0.8] | 0.42 |
| | gamma | [0, 20] | 0.6 |

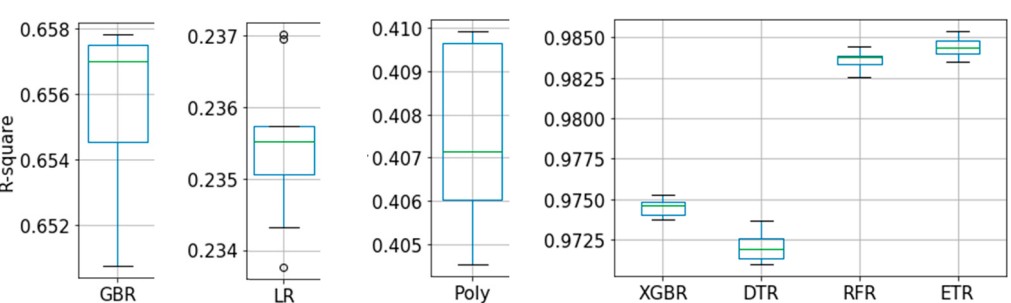

**Figure 5.** Box plot of R$^2$ obtained from different models in 10-fold cross-validation.

Figure 5 shows that most of the machine learning models, except GBR, delivered a good result, with a mean/median $R^2$ of over 97%. The reason for the low accuracy of GBR is that it is very sensitive to noise and hyperparameters compared to other methods, which results in its failure to generate a generalized model for the test dataset by overfitting the actual training data with high nonlinearity. ETR gave the most accurate result, closely followed by RFR. Table 7 shows the descriptive statistics of the models with total computational time. From the four models, DTR had the least computational time, showing that DTR-based ensemble models improve the accuracy of the single model. Relatively, GBR, LR, and polynomial regression models had lower accuracies than other models. As for the ensemble techniques, bagging provided a better result than boosting, but a single regressor (DTR) still provided a better mean $R^2$ with a slight increase in the variance. The DTR has an acceptable accuracy with a low computational time, but it has larger variability in the performance of the estimated model

than other boosting and bagging models. Accordingly, the DTR is not recommended for prediction of the SOG.

**Table 7.** Descriptive statistics of model accuracy in 10-fold cross-validation.

|  | LR | Poly | GBR | XGBR | DTR | RFR | ETR |
|---|---|---|---|---|---|---|---|
| Mean [%] | 23.55 | 40.76 | 65.59 | 97.459 | 97.209 | 98.37 | 98.45 |
| Std. [%] | 0.102 | 0.202 | 0.2357 | 0.056 | 0.0866 | 0.055 | 0.057 |
| Min [%] | 23.38 | 40.45 | 65.07 | 97.37 | 97.10 | 98.26 | 98.36 |
| Median [%] | 23.55 | 40.72 | 65.70 | 97.46 | 97.19 | 98.38 | 98.44 |
| Max [%] | 23.70 | 40.99 | 65.78 | 97.53 | 97.37 | 98.45 | 98.54 |
| Computational time [sec] | 8 | 850 | 6880 | 1514 | 312 | 2804 | 1590 |

A further assessment of the performance of a model is through its achieved accuracies using the testing dataset. From Table 8, it is observed that ETR performed slightly better than RFRs. In addition, the computational time was almost half of that of the RFR. Furthermore, the $R^2$ of XGBR and DTR even increased to approximately 96.98% and 96.46%, respectively. Finally, it is important to note that the investigation was executed using a computer with the following specifications: Windows 10 with 64-bit Operating System and x64-based processor, Intel(R) Xeon(R) CPU E3-123 v3 @3.30GHz processor, and 32.0 GB installed memory (RAM). By far, the computational time of DTR is better than the others after LR.

**Table 8.** Model performance for testing dataset.

| Model | $R^2$ | RMSE | NRMSE | Computational time [sec] |
|---|---|---|---|---|
| GBR | 0.6858 | 1.0608 | 0.0617 | 908 |
| XGBR | 0.9698 | 0.3287 | 0.0191 | 257 |
| DTR | 0.9646 | 0.3559 | 0.0207 | 52 |
| RFR | 0.9831 | 0.2464 | 0.0143 | 489 |
| ETR | 0.9847 | 0.2340 | 0.0136 | 253 |
| LR | 0.2379 | 1.6522 | 0.0961 | 1 |
| 3$^{rd}$ order Polynomial | 0.4008 | 1.4778 | 0.0859 | 120 |

In general, the ETR model has shown better accuracy than the other models. In addition, its computational time is relatively acceptable. To validate the consistency of the ETR model for different ship routes and ship types, it was tested by extracting different ship data as a testing dataset. As shown in Table 9, the performance of the ETR model was consistent for two tankers and three cargo ships with various ship routes, and thus, the ETR model is still valid for predicting the SOG for various ship data.

**Table 9.** Extra trees regressor (ETR) models performance for a single route of different vessels.

| Vessel Name | Vessel Type | Route | $R^2$ | RMSE | NRMSE | Data Size |
|---|---|---|---|---|---|---|
| A | Tanker | Chiba, JP to Townsville, AUS | 0.9845 | 0.0609 | 0.0077 | 2644 |
| B | Tanker | Burnie, AUS to Yokkaichi, JP | 0.9734 | 0.1506 | 0.0206 | 3351 |
| C | Cargo | Shibushi to Vancouver, CAN | 0.9827 | 0.1178 | 0.0127 | 4481 |
| D | Cargo | Marsden PT. to Singapore | 0.9881 | 0.0543 | 0.0106 | 3004 |
| E | Cargo | Westshore CAN, to Gwangyang. S. KOR | 0.9821 | 0.1038 | 0.0127 | 3014 |

## 4. Conclusions

This study proposed a data-driven methodology for the prediction of the SOG of a ship while sailing using the AIS data and noon-report marine weather data. The main findings of this study are as follows: The developed models can accurately estimate the SOG of the ships sailing under different weather conditions, load conditions, draughts, and sailing distance/direction; the results also showed that linear regression and the polynomial model gives inaccurate prediction results for SOG because

of the highly nonlinear tendency of SOG with time; using noon-report weather data and AIS data, various ensemble models achieved model accuracies of more than 96% as given by the $R^2$ value even considering the random effects of SOG; applying hyperparameter optimization may also increase and stabilize the accuracy of a model; and ETR, which is one of the bagging ensemble models, yielded high accuracy with low computational time to predict the SOG.

The suggested methodology was used for the real data with different ship types and routes, proving that this methodology can be applied to essentially for any types of vessel. In addition, while the findings of this study are expected to be used for route optimization purposes, this methodology can also be used to create models that will help the performance degradation of track vessels, and the optimization of shipping operations.

**Author Contributions:** Conceptualization, M.A., Y.N. and Y.S.; data acquisition, Y.S., S.L., and Y.N.; methodology, M.A.; coding, M.A. and Y.S.; validation, M.A.; formal analysis, M.A.; investigation, M.A.; resources, Y.N. and I.L.; writing-original draft preparation, M.A.; writing-review and editing, Y.N. All authors have read and agreed to the published version of the manuscript.

**Funding:** This research received no external funding.

**Acknowledgments:** This work was supported by the National Research Foundation of Korea (NRF) grant funded by the Korea government (MSIP) through GCRC-SOP (No. 2011-0030013), Korea government (MSIT) (No. 2018R1D1A1A02086093), and National Innovation Cluster Program (P0006887, Build on Cloud Intelligence Platform based Marine Data) funded by the Ministry of Trade, Industry & Energy (MOTIE) and Korea Institute for Advancement of Technology (KIAT).

**Conflicts of Interest:** The authors declare no conflict interest.

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
