# Peer review of "Machine Learning Approaches for Ship Speed Prediction towards Energy Efficient Shipping"

_applsci, doi:10.3390/app10072325_

Round 1

Reviewer 1 Report

The paper is very interesting and contemporary. Please, explain better how weather data (sea, swell, wind) are acquired. This is very important part of the input data.  

Author Response

Point 1: Please, explain better how weather data (sea, swell, wind) are acquired. This is very important part of the input data.

 Response 1: Dear reviewer, thank you for your valuable comment and suggestion, and in accord of your comment, an additional explanation of the noon-report weather data has been incorporated in the revised manuscript as follows.

“A 2018 AIS satellite data and noon-report weather data of 14 tankers and 62 cargo ships were collected. The AIS data and noon-report marine weather data was provided by Lab021, and the AIS data collected within an average of 3 min time intervals. The resolution of the weather data is 0.5 degree in the latitude and longitude directions.”

Reviewer 2 Report

This is a very high quality research. Authors have addressed everything. 

One more question, how do you think the real-time feasibility of deep learning models in practical applications.

Author Response

Point 1: how do you think the real-time feasibility of deep learning models in practical applications?

Response 1: Dear reviewer, first of all I would like to thank you for the comments and your time. To generate a deep learning model using real-time data, large amounts of data must be processed and modelled in a short period of time. Therefore, it is a common fact that using a deep learning model in real-world problems is not easy. However, our SOG model is generated by learning sufficient data (14 tankers and 62 cargo ships) with high accuracy and robustness even though the data are not the real time data. If the ship route optimization combined with the SOG model is performed at regular intervals, the route can be updated depending on the weather and operating conditions, but the SOG model is primarily determined from the data already learned and therefore the model update is unnecessary in real time. (Of course, the proposed SOG model can be updated using real-time data and the real-time data can be used as a database for further updating the model later.) The ship route optimization algorithm combined with the SOG model we have recently developed takes only a few minutes to find an economic ship route, so it is more practical than the one using real-time data. However, the proposed deep learning model is limitedly available for route optimization rather than being able to be used in general problem.

Reviewer 3 Report

  1. Affiliation shall include the complete address, i.e. name of the institution, street, city, code
  2. Improve the description, details and processing procedure of AIS data.
    1. Are the AIS data gathered by ground station or satellite?
    2. Which is the time span between successive acquisition? (about 1h for S-AIS, or minutes)
  3. Please, check Figure 3 and related description. If I understand well, there are outlier values which are different from the values typically used for not-available data (511 for heading and 1023 for velocity [1]). The outliers are identified and excluded from the processing using a threshold-based approach? Which is the failure rate of this approach?

[1]        M. D. Graziano, A. Renga, and A. Moccia, “Integration of Automatic Identification System (AIS) data and single-channel Synthetic Aperture Radar (SAR) images by sar-based ship velocity estimation for maritime situational awareness,” Remote Sens., vol. 11, no. 19, 2019.

Author Response

Response 1: Dear reviewer, thank you for your valuable comments and suggestions, the author affiliations are edited based on your suggestion and journal format as follows.

Misganaw Abebe 1, Yongwoo Shin2, Yoojeong Noh2, * Sangbong Lee3, Inwon Lee4

Research Institute of Mechanical Technology, Pusan Nat’l Univ., Busan, 46241, Rep. of Korea; [email protected]

2   School of Mechanical Engineering, Pusan Nat’l Univ., Busan, 46241, Rep. of Korea; [email protected]

3   Lab021, Busan, 48508, Rep. of Korea; [email protected]

4   Department of Naval Architecture & Ocean Engineering, Pusan Nat’l Univ., Busan, 46241, Rep. of Korea; [email protected]

Point 2: Improve the description, details and processing procedure of AIS data.

  1. Are the AIS data gathered by ground station or satellite?
  2. Which is the time span between successive acquisitions? (about 1h for S-AIS, or minutes)Response 2: Following your comment, additional explanation has been incorporated in the revised manuscript on AIS satellite data acquisition as follows.

Response 2: Following your comment, additional explanation has been incorporated in the revised manuscript on AIS satellite data acquisition as follows.

“A 2018 AIS satellite data and noon-report weather data of 14 tankers and 62 cargo ships were collected. The AIS data and noon-report marine weather data was provided by Lab021, and the AIS data collected within an average of 3 min time intervals. The resolution of the weather data is 0.5 degree in the latitude and longitude directions.”

Point 3: Please, check Figure 3 and related description. If I understand well, there are outlier values which are different from the values typically used for not-available data (511 for heading and 1023 for velocity [1]). The outliers are identified and excluded from the processing using a threshold-based approach? Which is the failure rate of this approach?

[1] M. D. Graziano, A. Renga, and A. Moccia, “Integration of Automatic Identification System (AIS) data and single-channel Synthetic Aperture Radar (SAR) images by sar-based ship velocity estimation for maritime situational awareness,” Remote Sens., vol. 11, no. 19, 2019.

Response 3: As you mentioned, AIS data has a default outlier value for not-available (missed data): such as 102.2 for SOG, 511 for heading, 91 for latitude, and 181 for longitude. We added the default outlier values in the revised paper including the reference [20] as follows.

“From the AIS data report [19], if the data value is not-available (missed data), there is a default outlier value for each feature such as 102.2 for SOG, 511 for heading, 91 for latitude and 181 for longitude [20]. Those values are observed in our data and used to discard the missed data.”

In addition to this default outlier value, we also observed a few odd values (noise data) in the scatter plot. Thus, to exclude those data before developing the prediction model, the authors used z-score with a high threshold value.

In the previous manuscript, we checked the scatter plot including the default outliers, but we checked again all features of the defined outlier values and discarded those data according to your comment. Then, we applied z-score for all feature to extract the extreme observed outlier data. To explain the modified part, additional sentences have been incorporated in the revised manuscript, and Figures 2 and 3 were also modified after the default outlier values are removed.

Round 2

Reviewer 3 Report

No further comments on my side. I suggest to accept the paper in the present form.